# Thin Film Flow of Micropolar Fluid in a Permeable Medium

**Vakkar Ali [1], Taza Gul [2], Shakeela Afridi [2], Farhad Ali [2], Sayer Obaid Alharbi [3] and Ilyas Khan [4,*]**

[1] Department of Mechanical and Industrial Engineering, Majmaah University, Al Majmaah 11952, Saudi Arabia; w.ahmad@mu.edu.sa
[2] Department of mathematics, City University of Science and Information Technology (CUSIT), Peshawar 25000, Pakistan; tazatazagul@cusit.edu.pk (T.G.); shakeelaafridi@gmail.com (S.A.); farhadaliecomaths@yahoo.com (F.A.)
[3] Department of Mathematics, College of Science Al-Zulfi, Majmaah University, Al-Majmaah 11952, Saudi Arabia; so.alharbi@mu.edu.sa
[4] Faculty of Mathematics and Statistics, Ton Duc Thang University, Ho Chi Minh City, Vietnam
[*] Correspondence: ilyaskhan@tdt.edu.vn

**Abstract:** The thin film flow of micropolar fluid in a porous medium under the influence of thermophoresis with the heat effect past a stretching plate is analyzed. Micropolar fluid is assumed as a base fluid and the plate is considered to move with a linear velocity and subject to the variation of the reference temperature and concentration. The latitude of flow is limited to being two-dimensional and is steadily affected by sensitive fluid film size with the effect of thermal radiation. The basic equations of fluid flow are changed through the similarity variables into a set of nonlinear coupled differential equations with physical conditions. The suitable transformations for the energy equation is used and the non-dimensional form of the temperature field are different from the published work. The problem is solved by using Homotopy Analysis Method (HAM). The effects of radiation parameter $R$, vortex-viscosity parameter $\Delta$, permeability parameter $Mr$, microrotation parameter $Gr$, Soret number $Sr$, thermophoretic parameter $\tau$, inertia parameter $Nr$, Schmidt number $Sc$, and Prandtl number $Pr$ are shown graphically and discussed.

**Keywords:** thin film of micropolar fluid; porous medium; thermophoresis; thermal radiation; skin friction; Nusselt number and Sherwood number; variable thickness of the liquid film; HAM

---

## 1. Introduction

Fluids, generally, have a major role in many problems related to industrial and engineering applications like crystal growing, glass blowing, polymer extrusion processes, metallurgical processes, and so on. In the extrusion process, the heated liquid stretching into a cooling system, as well as the phenomenon in which the tiny sized particles are transferred from a hot surface to a cool surface, is called thermophoresis. In gasses, tiny particles like dust exert force parallel to the temperature gradient called thermophoretic force, and the motion gained by these particles is known as thermophoretic velocity. In thermophoresis, tiny particles are transferred towards cold surfaces, whereas hot surface particles also resist taking place and, as a result, a particle free layer is observed around the hot surface, as analyzed by Goldsmith and May [1]. The most important application of this phenomenon is to remove tiny particles from the path of gas particles used in turbine blades. The same phenomenon was used by Goren [2] in the study of aerosol particles, and this idea was extended by Jayaraj et al. [3] in the natural convection. The idea of mass transfer in this phenomenon was investigated by Selim et al. [4]. They analyzed the effects of physical parameters involved in the model.

Chamka et al. [5,6] observed the thermophoresis effect in free convection boundary layer flow over the permeable wall. Das [7] studied variable fluid properties with slip boundary conditions. Flow in porous media is highly important in enhanced oil recovery, geothermal energy extraction, insulation of buildings, food processing, heat storage beds, composite manufacturing, and the coating of paper and textile processes. Porous media flow describes different practical and engineering applications like oil or gaseous movement, liquid in the oil reservoir or gaseous field, the purification process of oil, gaseous wells, drilling, and the processing of carbon made substances and cosmetic material.

Generally, the study of non-Newtonian fluid flow in two- and three-dimensional problems is a tough job because of its high nonlinearity and, especially, the addition of extra terminologies such as magnetic field, porous medium, thermophoretic term, dissipation term, and so on. Despite these difficulties, efforts are being made by the researchers to solve such problems. The idea of viscous dissipation and permeable media was introduced by Al-Hadrami et al. [8]. In another paper, Al-Hadrami et al. [9] studied the combined problem of convection for both forced and free convection through a permeable channel. The micropolar fluids in two and three dimensions belong to the non-Newtonian fluids explained by Łukaszewicz [10] in his book. It is pointed out that the Navier-Stokes equation is not sufficient to handle the Cauchy stress tensor of micropolar fluid and, therefore, this fluid belongs to non-Newtonian fluids. Aouadi [11] presented a numerical solution for micropolar liquid flow over a stretched plate. The flow of second grade fluid with heat flux over a stretching surface is described in the studies of Chauhan and Olkha [12] and Cortell [13]. Dandapat and Gupta [14] observed the allied problem over a stretching sheet with some modification. The time-dependent motion of second order liquid in partially filled porous media was explored by Chuhan and Kumar [15]. Khan and Shafie [16] studied the generalized Burger's fluid including rotation in a porous medium. They observed effects of embedded parameters related to the model. Micropolar fluid is one of the important sub-class of non-Newtonian fluid. Studies related to micropolar fluids with various physical configurations with thermal radiation were presented by Abo-Eldahab and Ghonaim [17], Rashidi et al. [18,19], Heydari et al. [20], and Tripathy et al. [21]. The idea of heat and mass transfer mechanisms were described by the researchers to study the impact of various embedded parameters on the nanoparticle volume fraction. Rahman and Sattar [22] and Bakr [23] have studied the heat and mass transfer flow of micropolar fluid using the oscillatory boundary conditions. Ramzan et al. [24] have examined the Buoyancy impacts on the heat and mass transfer flow of the micropolar fluid with double stratification. Srinivasacharya and Ramreddy [25] have inspected the heat and mass transfer in micropolar fluid with thermal and mass stratification.

Recently, thin film flow has been an important subject of research. Thin film fluid is used for making different heat exchangers and tools in chemical techniques, and these applications require complete comprehension on the motion procedure. The applications comprise wire and fiber coating, polymer preparing, and so on. This motion is attached to manufacturing various types of sheets, either metallic or plastic. The quality of the final product is related to heat and mass transport and the rate of stretching. An analysis of heat transfer in Williamson nanofluid flow was conducted by Nadeem and Hussain [26] and Khan et al. [27]. Aziz et al. [28] studied heat transfer through thin film flow on an unsteady stretching sheet with internal heating. Qasim et al. [29] and Tawade et al. [30] discussed the flow of thin film using different fluids and geometries. Khan et al. [31] and Mahmood and Khan [32] investigated the effects of different variables on different fluids in their flow. According to our knowledge, there is no published work related to thermophoresis on heat transfer and thermal radiation characteristics of thin film micropolar liquid on the stretched plate under the transformations used in this research. Therefore, we have shown our interest in this paper to make an effort in discussing this new case. In this manuscript, exploration of the behavior of a steady, laminar, and two-dimensional flow of an incompressible micropolar fluid thin film into a porous medium past a stretched sheet was examined. Further, the inclusion of thermal radiation in the equation of energy is always used as a special case and, in most of the problems in the existing literature, the energy equation is used without radiation. In the papers cited above [17–20], the non-dimensional energy equation is written

as $(3R + 4)\theta'' + 3R \, Pr \, f\theta' = 0$, in which $R$ is revealed as the radiation term. Clearly, if $R$ becomes zero, then the energy equation is reduced to $\theta'' = 0$, that is, the key parameter $Pr$ and momentum boundary layer vanish and, therefore, the energy equation becomes meaningless. Therefore, we have tried to avoid this situation by using a transformation that is the same as in the works of [27,29,32] for the same problem as cited in the literature [17–20] with the addition of concentration. In recent research, most researchers used homotopy analysis method (HAM) to solve higher order nonlinear problems, and credit goes to Liao [33–35], who investigated such a wonderful technique to solve nonlinear higher order differential equations. Gul et al. [36,37] used the HAM method for the suitable range of parameters. Analytical solutions in series form are calculated using HAM. The effects of all parameters on velocity, microrotation, temperature, and concentration fields are shown graphically.

## 2. Mathematical Formulation

Consider the thin film micropolar fluid flow on a stretched plate, which is being stretched with a linear velocity $U_w = ax$. Here, $a > 0$ is a constant and shows the stretching rate and $x$ displays the direction of the flow. The thickness δ of the thin film is chosen uniform and the medium is considered porous, as displayed in Figure 1. The stretching plate is kept at temperature $T_w$ and concentration $C_w$. The temperature $T_w = T_0 - T_{ref}\left(\frac{U_w x}{2v}\right)$ and concentration $C_w = C_0 - C_{ref}\left(\frac{U_w x}{2v}\right)$ on the surface are assumed to vary with distance $x$ from the plate. $T_0$ and $C_0$ are the temperature and concentration at the plate, while $T_{ref}$ and $C_{ref}$ are the constant reference temperature and concentration. Further, it is assumed that the liquid film is gripping and releasing radiation. The radiate heat flux is considered along the *x*-axis, while neglecting along the *y*-axis.

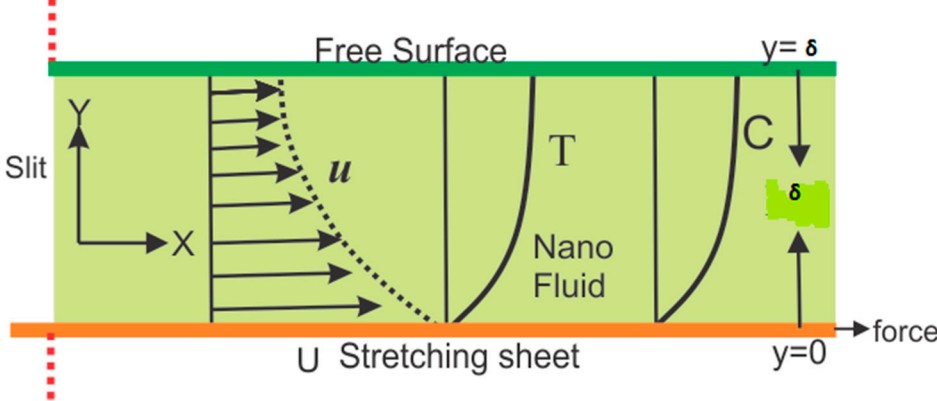

**Figure 1.** Physical geometry of the problem.

The basic flow equations of our proposed model are as follows:

$$u_x + v_y = 0 \tag{1}$$

$$uu_x + vu_y = vu_{xx} + k_c\sigma_y + \frac{v\varphi}{K}(U - u) + C_r\varphi(U^2 - u^2) \tag{2}$$

$$G_1\sigma_{yy} - 2\sigma - u_y = 0 \tag{3}$$

$$\rho c_p\left(uT_x + vT_y\right) = kT_{yy} - (q_r)_y \tag{4}$$

$$uC_x + vC_y = D_mC_{yy} + \frac{D_m k_T}{T_m}T_{yy} - (V_TC)_y \tag{5}$$

The modeled boundary conditions for the two-dimensional liquid film are as follows:

$$u = U_w = ax, \, v = 0, \, \sigma = 0, \, T = T_w, \, C = C_w \text{at} y = 0, \tag{6}$$

$$u_y = T_y = \sigma_y = C_y = 0, \ v = \delta_x, \text{at} \, y = \delta. \tag{7}$$

The Rosseland approximation is defined as follows:

$$q_r = -\frac{4\sigma^*}{3k^*}\partial_y T^4 \tag{8}$$

where $q_r$ is radiative heat flux, $\sigma^*$ is Stefan–Boltzman constant, and $k^*$ represents the mean absorption coefficient. The flux is assumed to be small, such that $T_1^5$ and higher terms are ignored, as in the existing literature. After expanding by Taylor's series, $T^4$ is reduced to the following form:

$$T^4 = 4T_1^3 T - 3T_1^4 \tag{9}$$

$T_1$ is used as the temperature at the free surface. Using Equations (8) and (9), Equation (4) is reduced as follows:

$$uT_x + vT_y = \frac{k}{\rho c_p}T_{yy} + \frac{16\sigma^* T_1^3}{3\rho c_p k^*}T_{yy} \tag{10}$$

Abo-Eldahab and Ghonaim [17], Rashidi et al. [18,19] and Heydari et al. [20] introduced the following transformations:

$$\psi(x,y) = (2vU_w x)^{\frac{1}{2}}f(\eta), \ \sigma = \left(\frac{U_w}{2vx}\right)^{\frac{1}{2}}U_w g(\eta), \ \eta = \left(\frac{U_w}{2vx}\right)^{\frac{1}{2}}y, \ u_x = \psi_y \text{ and } u_y = -\psi_x \tag{11}$$

In the recent research of Khan [27] and Qasim et al. [29], the thin film flows are modeled using reference temperature and concentration for steady and unsteady problems, respectively.

$$T = T_0 - T_{ref}\left(\frac{U_w x}{2v}\right)\theta(\eta), \ C = C_0 - C_{ref}\left(\frac{U_w x}{2v}\right)\theta(\eta) \tag{12}$$

where $T_0$ is temperature at the stretched surface and $T_{ref}$ is used as a constant reference temperature, such that $0 \leq T_{ref} \leq T_0$. Similarly, $C_0$ is the concentration at the stretched surface and $C_{ref}$ is used as a constant reference concentration, such that $0 \leq C_{ref} \leq C_0$. Substituting Equations (11) and (12) into Equations (1)–(7), the basic governing equations of velocity, velocity rotation, and temperature with boundary conditions yield the following forms:

$$f''' + ff'' + \Delta g' + \frac{1}{Mr}\left(1 - f'\right) + Nr\left(1 - (f')^2\right) = 0 \tag{13}$$

$$Gr \, g'' - 2(2g + f'') = 0 \tag{14}$$

$$\left(1 + \frac{4}{3}R\right)\theta'' - Pr\left(2\theta f' - f\theta'\right) = 0 \tag{15}$$

$$\phi'' + Sc(Sr - \tau\phi)\theta'' + Sr\left(f - \tau\theta'\right)\phi' - 2Sc\phi f' = 0 \tag{16}$$

$$f(0) = g(0) = 0, \ f'(0) = \theta(0) = \phi(0) = 1 \tag{17}$$

$$f''(\beta) = f(\beta) = g'(\beta) = \theta'(\beta) = \phi'(\beta) = 0 \tag{18}$$

where $f$ is a dimensionless velocity function and $g$ is a dimensionless microrotation angular velocity function, $\theta$ is the temperature function, $\phi$ is the concentration function, $\beta$ is the non-dimensional thickness of the liquid film, $\Delta = \frac{k_1}{v}$ is the vortex–viscosity parameter, $Mr = \frac{Ka}{2\phi v}$ is the permeability parameter, $Nr = \frac{2\phi C_r U_w}{a}$ is the inertia coefficient parameter, $Gr = \frac{G_1 a}{v}$ represents the microrotation parameter, $Pr = \frac{\rho v c_p}{k}$ represents the Prandtl number, $R = \frac{4\sigma^* T_1^3}{k^* k}$ represents the radiation parameter, $Sc = \frac{v}{D_m}$ represents the Schmidt number, $Sr = \frac{D_m k_T (T_w - T_0)}{vT_m(C_w - C_0)}$ represents the Soret number, and $\tau = \frac{kU_w^2}{2va}$ is the thermophoretic parameter and is same as in the works of [17–20].

The important physical quantities are skin friction coefficient $C_f$, local Nusselt number $Nu$, and Sherwood number, which are defined as follows:

$$C_f = \frac{\mu(u_y)_{y=0}}{\frac{1}{2}\rho U_w^2}, Nu = \frac{-k(T_y)_{y=0}x}{k(T_w - T_0)}, Sh = \frac{-D_m(C_y)_{y=0}x}{D_m(C_w - C_0)}.$$

where $\mu(u_y)_{y=0}$, $-k(T_y)_{y=0}$, and $-D_m(C_y)_{y=0}$ are shear stress, heat, and mass fluxes at the surface, respectively. Using the variables in (11), the expressions for dimensionless skin friction, Nusselt number, and Sherwood number are obtained as follows:

$$C_f \left(\frac{\text{Re}}{2}\right)^{\frac{1}{2}} = -f''(0), Nu \left(\frac{\text{Re}}{2}\right)^{-\frac{3}{2}} = -\theta'(0), Sh \left(\frac{\text{Re}}{2}\right)^{-\frac{3}{2}} = -\phi'(0) \qquad (19)$$

Here, $\text{Re} = \frac{U_w x}{v}$ represents the Reynold number based on the stretching velocity. The calculated values for the skin friction coefficient and local Nusselt number are shown in Tables 1–3.

**Table 1.** Values for the skin friction coefficient, when $h = -0.2, Mr = Gr = 0.8, Nr = R = \Delta = Sc = Sr = \tau = Pr = 0.3, \beta = 1$.

| Δ | *Mr* | *Nr* | $-f''(0)$ |
|---|---|---|---|
| 0.3 | 0.8 | 0.3 | 1.36594 |
| 0.4 | 0.8 | 0.3 | 1.36571 |
| 0.5 | 0.8 | 0.3 | 1.36547 |
| 0.3 | 0.8 | 0.3 | 1.36594 |
| 0.3 | 0.9 | 0.3 | 1.24938 |
| 0.3 | 1.0 | 0.3 | 1.15533 |
| 0.3 | 0.8 | 0.3 | 1.36594 |
| 0.3 | 0.8 | 0.4 | 1.45338 |
| 0.3 | 0.8 | 0.5 | 1.54067 |

**Table 2.** Values of rate of heat transfer or the local Nusselt number, when $h = -0.2, Mr = Gr = 0.8, Nr = R = \Delta = Sc = Sr = \tau = Pr = 0.3, \beta = 1$.

| *R* | *Pr* | $-\theta'(0)$ |
|---|---|---|
| 0.3 | 0.3 | 0.246741 |
| 0.4 | 0.3 | 0.240841 |
| 0.5 | 0.3 | 0.235105 |
| 0.3 | 0.3 | 0.246741 |
| 0.3 | 0.4 | 0.325885 |
| 0.3 | 0.5 | 0.403524 |

**Table 3.** Values of the Sherwood number, when $h = -0.2, Mr = Gr = 0.8, Nr = R = \Delta = Sc = Sr = \tau = Pr = 0.3, \beta = 1$.

| *Sc* | *Sr* | *τ* | $-\phi'(0)$ |
|---|---|---|---|
| 0.3 | 0.3 | 0.3 | 0.265463 |
| 0.4 | 0.3 | 0.3 | 0.350588 |
| 0.5 | 0.3 | 0.3 | 0.434081 |
| 0.3 | 0.3 | 0.3 | 0.265463 |
| 0.3 | 0.4 | 0.3 | 0.264059 |
| 0.3 | 0.5 | 0.3 | 0.262655 |
| 0.3 | 0.3 | 0.3 | 0.265463 |
| 0.3 | 0.3 | 0.4 | 0.266868 |
| 0.3 | 0.3 | 0.5 | 0.268272 |

## 3. Solution Methodology

### 3.1. Homotopy Analysis Method

The solutions of Equations (13)–(16) with the related boundary conditions (17) and (18) are achieved using HAM. Consider that initial guesses on $f(\eta)$, $g(\eta)$, $\theta(\eta)$, and $\phi(\eta)$ satisfying the boundary conditions at $\eta = 0$ are as follows:

$$f_0(\eta) = \frac{\eta^3}{2\beta^2} - \frac{3\eta^2}{2\beta} + \eta, \; g_0(\eta) = 0, \theta_0(\eta) = 1, \; \phi_0(\eta) = 1 \tag{20}$$

The linear operators for the given functions are the following:

$$L_f(f) = f^{(iv)}, \; L_g(g) = g'', \; L_\theta(\theta) = \theta'', \; L_\phi(\phi) = \phi''. \tag{21}$$

satisfying the following properties:

$$L_f\left(a_1 + a_2\eta + a_3\eta^2 + a_4\eta^3\right) = 0, \; L_g(a_5 + a_6\eta) = 0, \; L_\theta(a_7 + a_8\eta) = 0, \; L_\phi(a_9 + a_{10}\eta) = 0 \tag{22}$$

where $a_i(i = 1 - 10)$ are constants related to the general solution.

The corresponding nonlinear operators are as follows:

$$\begin{aligned} N_f[f(\eta;q), g(\eta;q)] = \quad & f_{\eta\eta\eta}(\eta;q) + f(\eta;q)f_{\eta\eta}(\eta;q) + \Delta g_\eta(\eta;q) \\ & + \frac{1}{Mr}(1 - f_\eta(\eta;q)) + Nr\left(1 - (f_\eta(\eta;q))^2\right), \end{aligned} \tag{23}$$

$$N_g[f(\eta;q), g(\eta;q)] = Grg_{\eta\eta}(\eta;q) - 2(2g(\eta;q) + f_{\eta\eta}(\eta;q)) = 0, \tag{24}$$

$$N_\theta[f(\eta;q), \theta(\eta;q)] = \left(1 + \frac{4}{3}R\right)\theta_{\eta\eta}(\eta;q) - Pr(2\theta(\eta;q)f_\eta(\eta;q) - f(\eta;q)\theta_\eta(\eta;q)), \tag{25}$$

$$\begin{aligned} N_\phi[f(\eta;q), \theta(\eta;q), \phi(\eta;q)] = \quad & \phi_{\eta\eta}(\eta;q) + Sc(Sr - \tau\phi(\eta;q))\,\theta_{\eta\eta}(\eta;q) + \\ & Sr(f - \tau\theta_\eta(\eta;q))\phi_\eta(\eta;q) - 2Sc\phi(\eta;q)f_\eta(\eta;q) = 0. \end{aligned} \tag{26}$$

(a)  Zeroth-Order Deformation Problem

The main idea of HAM is explained in Equations (19)–(22). We formulate the zeroth-order problem from Equations (13)–(16) as follows:

$$(1 - q)L_f\{f(\eta;q) - f_0(\eta)\} = qh_f N_f\{f(\eta;q), g(\eta;q)\}, \tag{27}$$

$$(1 - q)L_g\{g(\eta;q) - g_0(\eta)\} = qh_g N_g\{f(\eta;q), g(\eta;q)\}, \tag{28}$$

$$(1 - q)L_\theta\{\theta(\eta;q) - \theta_0(\eta)\} = qh_\theta N_\theta\{f(\eta;q), \theta(\eta;q)\}, \tag{29}$$

$$(1 - q)L_\phi\{\phi(\eta;q) - \phi_0(\eta)\} = qh_\phi N_\phi\{f(\eta;q), g(\eta;q), \theta(\eta;q), \phi(\eta;q)\}, \tag{30}$$

Expanding the functions $f$, $g$, $\theta$ and $\phi$ by Taylor's series when $q = 0$, we have the following:

$$\begin{aligned} f(\eta;q) &= f_0(\eta) + \sum_{w=1}^{\infty} f_w(\eta)\, q^w, \\ g(\eta;q) &= g_0(\eta) + \sum_{w=1}^{\infty} g_w(\eta)\, q^w, \\ \theta(\eta;q) &= \theta_0(\eta) + \sum_{w=1}^{\infty} \theta_w(\eta)\, q^w, \\ \phi(\eta;q) &= \phi_0(\eta) + \sum_{w=1}^{\infty} \phi_w(\eta)\, q^w. \end{aligned} \tag{31}$$

where

$$f_w(\eta) = \frac{1}{w!} f_\eta^w(\eta; q)|_{q=0}, \; g_w(\eta) = \frac{1}{w!} g_\eta^w(\eta; q)|_{q=0},$$
$$\theta_w(\eta) = \frac{1}{w!} \theta_\eta^w(\eta; q)|_{q=0}, \; \phi_w(\eta) = \frac{1}{w!} \phi_\eta^w(\eta; q)|_{q=0}. \tag{32}$$

The supporting constraints $h_f, h_g, h_\theta$, and $h_\phi$ are taken such that series (33) converges at $q = 1$. Substituting $q = 1$ in (33) we get the following:

$$f(\eta) = f_0(\eta) + \sum_{w=1}^{\infty} f_w(\eta), \tag{33}$$

$$g(\eta) = g_0(\eta) + \sum_{w=1}^{\infty} g_w(\eta), \tag{34}$$

$$\theta(\eta) = \theta_0(\eta) + \sum_{w=1}^{\infty} \theta_w(\eta), \tag{35}$$

$$\phi(\eta) = \phi_0(\eta) + \sum_{w=1}^{\infty} \phi_w(\eta). \tag{36}$$

(b)  $w^{th}$ Order Deformation Problem

The following equations are satisfied by the problem of the $w^{th}$ order.

$$L_f[f_w(\eta) - \chi_w f_{w-1}(\eta)] = h_f \, R_w^f(\eta), \tag{37}$$

$$L_g[g_w(\eta) - \chi_w g_{w-1}(\eta)] = h_g \, R_w^g(\eta), \tag{38}$$

$$L_\theta[\theta_w(\eta) - \chi_w \theta_{w-1}(\eta)] = h_\theta \, R_w^\theta(\eta), \tag{39}$$

$$L_\phi[\phi_w(\eta) - \chi_w \phi_{w-1}(\eta)] = h_\phi \, R_w^\phi(\eta). \tag{40}$$

where

$$\chi_w = \begin{cases} 0, & \text{if } q \le 1 \\ 1, & \text{if } q > 1 \end{cases}$$

### 3.2. Numerical Solution

The numerical (ND solve) solution of Equations (13)–(16) with boundary conditions (17) and (18) for different values of embedded parameters are calculated and compared with HAM in Tables 4–7.

**Table 4.** Comparison of HAM and numerical solution for velocity when $h = -0.001, Pr = 10.6, Nr = R = Sc = Sr = \tau = \Delta = 0.5, Gr = \beta = Mr = 1.$

| $\eta$ | HAM Solution of $f'(\eta)$ | Numerical Solution | Absolute Error |
|---|---|---|---|
| 0 | $5.09 \times 10^{-22}$ | 0.000000 | $5.09 \times 10^{-22}$ |
| 0.1 | 0.099999 | 0.100043 | $4.3 \times 10^{-5}$ |
| 0.2 | 0.199999 | 0.200168 | $1.6 \times 10^{-4}$ |
| 0.3 | 0.299999 | 0.300364 | $3.6 \times 10^{-4}$ |
| 0.4 | 0.399999 | 0.400624 | $6.2 \times 10^{-4}$ |
| 0.5 | 0.499999 | 0.500937 | $9.3 \times 10^{-4}$ |
| 0.6 | 0.599999 | 0.601295 | $1.2 \times 10^{-3}$ |
| 0.7 | 0.699999 | 0.701689 | $1.6 \times 10^{-3}$ |
| 0.8 | 0.799999 | 0.802110 | $2.1 \times 10^{-3}$ |
| 0.9 | 0.899999 | 0.902549 | $2.5 \times 10^{-3}$ |
| 1 | 0.999999 | 1.002997 | $2.9 \times 10^{-3}$ |

**Table 5.** Comparison of HAM and numerical solution for microrotation angular velocity when $h = -0.15, Nr = R = \Delta = Sc = Sr = \tau = 0.5, Mr = \beta = 1, Gr = 5, Pr = 10.6$.

| η | HAM Solution of $g(\eta)$ | Numerical Solution | Absolute Error |
|---|---|---|---|
| 0 | −1.28767269 | −0.0000000 | $1.2 \times 10^{-8}$ |
| 0.1 | −0.0094987 | −0.0088208 | $6.7 \times 10^{-4}$ |
| 0.2 | −0.0167880 | −0.0157937 | $9.9 \times 10^{-4}$ |
| 0.3 | −0.0221826 | −0.0211439 | $1.2 \times 10^{-3}$ |
| 0.4 | −0.0259819 | −0.0250914 | $8.9 \times 10^{-4}$ |
| 0.5 | −0.0284720 | −0.0278531 | $6.1 \times 10^{-3}$ |
| 0.6 | −0.0299279 | −0.0296435 | $2.8 \times 10^{-4}$ |
| 0.7 | −0.0306156 | −0.0306741 | $5.8 \times 10^{-5}$ |
| 0.8 | −0.0307942 | −0.0311543 | $3.6 \times 10^{-4}$ |
| 0.9 | −0.0307176 | −0.0312916 | $5.7 \times 10^{-4}$ |
| 1 | −0.0306366 | −0.0312919 | $6.5 \times 10^{-4}$ |

**Table 6.** Comparison of HAM and numerical solutions for temperature when $h = -0.33, Nr = R = \Delta = Sc = Sr = \tau = 0.5, Mr = \beta = 1, Gr = 5, Pr = 10.6$.

| η | HAM Solution of $\theta(\eta)$ | Numerical Solution | Absolute Error |
|---|---|---|---|
| 0 | 0.999999989 | 1.000000 | $1.06 \times 10^{-8}$ |
| 0.1 | 0.924017 | 0.925128 | $1.1 \times 10^{-3}$ |
| 0.2 | 0.855763 | 0.857526 | $1.7 \times 10^{-3}$ |
| 0.3 | 0.795601 | 0.797495 | $1.8 \times 10^{-3}$ |
| 0.4 | 0.743742 | 0.745268 | $1.5 \times 10^{-3}$ |
| 0.5 | 0.700264 | 0.701015 | $7.5 \times 10^{-4}$ |
| 0.6 | 0.665136 | 0.664844 | $2.9 \times 10^{-4}$ |
| 0.7 | 0.638237 | 0.636805 | $1.4 \times 10^{-3}$ |
| 0.8 | 0.619370 | 0.616887 | $2.4 \times 10^{-3}$ |
| 0.9 | 0.608278 | 0.605027 | $3.2 \times 10^{-3}$ |
| 1 | 0.604656 | 0.601109 | $3.5 \times 10^{-3}$ |

**Table 7.** Comparison of HAM and numerical solutions for concentration when $h = -0.42, Nr = R = \Delta = Sc = Sr = \tau = 0.5, Mr = \beta = Gr = 1, Pr = 10.6$.

| η | HAM Solution of $\phi(\eta)$ | Numerical Solution | Absolute Error |
|---|---|---|---|
| 0 | 1.000000091 | 1.000000 | $9.1 \times 10^{-8}$ |
| 0.1 | 0.898855 | 0.90411 | $5.2 \times 10^{-3}$ |
| 0.2 | 0.809875 | 0.81792 | $8.04 \times 10^{-3}$ |
| 0.3 | 0.733047 | 0.741694 | $8.6 \times 10^{-3}$ |
| 0.4 | 0.668142 | 0.675623 | $7.4 \times 10^{-3}$ |
| 0.5 | 0.614767 | 0.619828 | $5.1 \times 10^{-3}$ |
| 0.6 | 0.572423 | 0.57436 | $1.9 \times 10^{-3}$ |
| 0.7 | 0.540544 | 0.539208 | $1.3 \times 10^{-3}$ |
| 0.8 | 0.518529 | 0.514298 | $4.2 \times 10^{-3}$ |
| 0.9 | 0.505765 | 0.499496 | $6.2 \times 10^{-3}$ |
| 1 | 0.501644 | 0.494614 | $7.03 \times 10^{-3}$ |

## 4. Graphical Results and Discussion

The thin film motion of a micropolar fluid through porous media with the impact of energy radiation and thermophoresis through a stretching plate is investigated. The non-linear coupled differential Equations (13)–(16) with physical conditions (17) and (18) were determined through HAM. The effects of all the embedded constants on the dimensionless velocity field, dimensionless microrotation, dimensionless temperature field, and concentration fields—$f(\eta)$, $g(\eta)$, $\theta(\eta)$, and $\phi(\eta)$, respectively—are observed. The physical geometry of the modeled problem is demonstrated by Figure 1. Liao [33–35] presented $h$ curves to measure the convergence of the series solution for accurate

results of the system, so suitable *h*-curves are drawn for the velocity profile $f(\eta)$, microrotation profile $g(\eta)$, temperature profile $\theta(\eta)$, and concentration profile $\phi(\eta)$ in range of $-2.0 \leq h_f \leq 0.1$, $-2 \leq h_g \leq 0$, $-2.1 \leq h_\theta \leq 0.1$, and $-2 \leq h_\phi \leq 0$, respectively, in Figures 2–5. The influence of permeability parameter *Mr* on the velocity field is described in Figure 6. The permeability parameter should be increased at a very small level because of the small thickness of the liquid film because higher values of *Mr*, that is , $Mr \rightarrow \infty$ correspond to the case in which there is no porous medium. The increasing values of *Mr* respond to the large opening of the porous space, which reduces retardation of the flow; so for increasing values of *Mr*, the velocity increases in this region. The larger values of the inertia coefficient parameter *Nr* increase the velocity of fluid as a result of its direct relation with fluid motion, deliberated in Figure 7. The influence of Δ versus motion of liquid film is represented in Figure 8. As Δ has an inverse relation with viscosity, the viscosity falls for larger values of Δ, while the velocity of the liquid film is raised. Figures 9 and 10 indicate the relationship between β with the fluid velocity profile $f(\eta)$ and microrotation profile $g(\eta)$. The fluid motion reduces with the increase in the liquid film thickness. The reason is clear, because larger values of β dominate the viscous forces and, as a result, the fluid velocity decreases. In other words, the thickness of the liquid film shows resistance to liquid flow, and fluid velocity causes retardation towards the free surface—this effect is very clear in the rotation velocity field $g(\eta)$. The microrotation profile $g(\eta)$ of the liquid film rises with the increasing microrotation *Gr*, as displayed in Figure 11, because the microrotation parameter has an inverse relation with the viscosity parameter. As a result, the viscosity reduces with the rising values of *Gr*; therefore, larger values of *Gr* offer low resistance to the flow and the velocity of fluid increases. Figure 12 demonstrates the variation of the inertia parameter *Nr* on the non-dimensional microrotation profile $g(\eta)$. It is observed that the rise in the inertia parameter *Nr* material parameter reduces the microrotation profile. The inclusion of thermal radiation in the equation of energy is always used as a special case and, in most of the problems in the existing literature, the energy equation is used without radiation. If the thermal radiation parameter *R* becomes zero, the temperature field $\theta(\eta)$ in Abo-Eldahab and Ghonaim [17], Rashidi et al. [18,19], and Heydari et al. [20] becomes meaningless, so it is not clear when the thermal radiation parameter *R* becomes zero in these papers. Therefore, our case of thermal radiation is reciprocal to the above published work, and is the same as Khan [27], Qasim et al. [29], and Mahmood and Khan [32]. Therefore, the temperature rises with the larger values of thermal radiation parameter, as shown in Figure 13, because the thickness of the boundary layer (thin film) is directly related to thermal radiation. Physically, the rate of energy transport increases and, as a result, the temperature of the fluid rises. The dimensionless fluid thickness β has a vital role in temperature distribution. $\theta(\eta)$ decreases with increasing values of β, which is obvious from Figure 14. The size of thin film absorbing heat, and thus the temperature of the fluid, decreases and, as a result, a cooling effect is produced. In other words, the thickness of the fluid decreases with the increasing temperature. Figure 15 represents the comparison of temperature and Prandtl number *Pr*. The temperature falls with growing values of *Pr*. In fact, the larger values of *Pr* enhance the viscous diffusion more than the thermal diffusion and, as a result, the temperature profile declines. Schmidt number verses concentration is deliberated in Figure 16. The rising values of Schmidt number *Sc* decrease the concentration field, because molecular diffusivity is inversely related to *Sc*. The contribution of the Soret number *Sr* is represented in Figure 17, showing that $\phi(\eta)$ rises when the Soret number *Sr* increases. In fact, the larger Soret number increases the viscosity and, therefore, $\phi(\eta)$ accelerates. Figure 18 shows the relationship between thermophoretic parameter τ and $\phi(\eta)$. They are inversely related to each other. Rising values of τ reduce the size of the boundary layer. The concentration field rises as thickness β increases, as shown in Figure 19, because of cohesive forces between molecules dominated by the increasing value of the parameter β, which result a rise in friction force and cause the fluid flow.

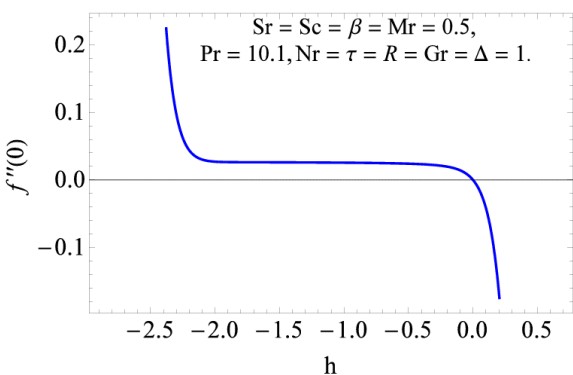

**Figure 2.** $h_f$ curves for the velocity field.

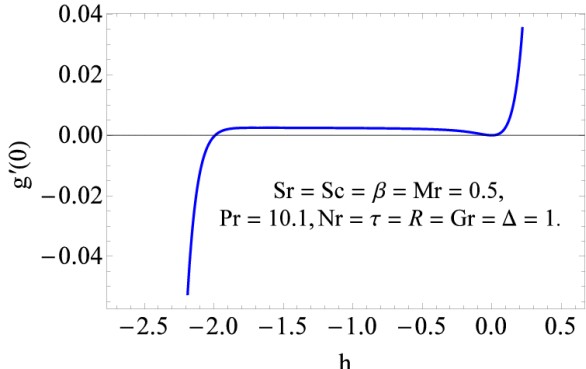

**Figure 3.** $h_g$ curves for the velocity field in rotation.

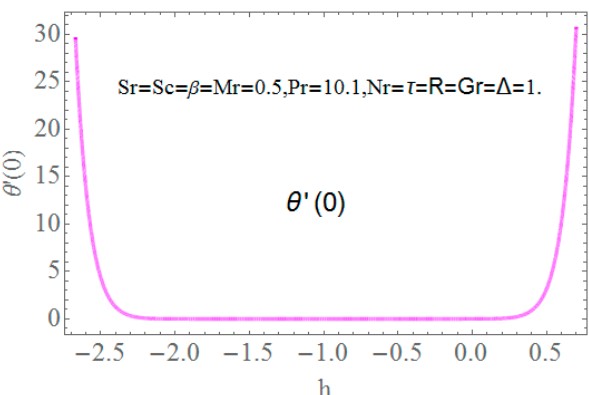

**Figure 4.** $h_\theta$ curves for the temperature field.

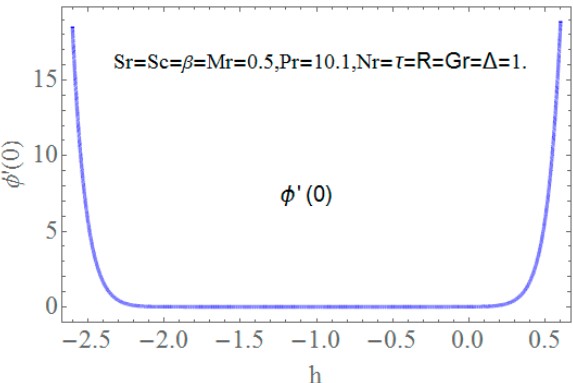

**Figure 5.** $h_\phi$ curves for the concentration field.

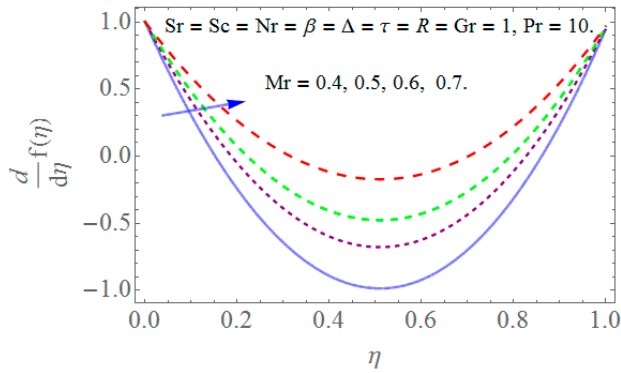

**Figure 6.** Effect of permeability parameter *Mr* on the velocity.

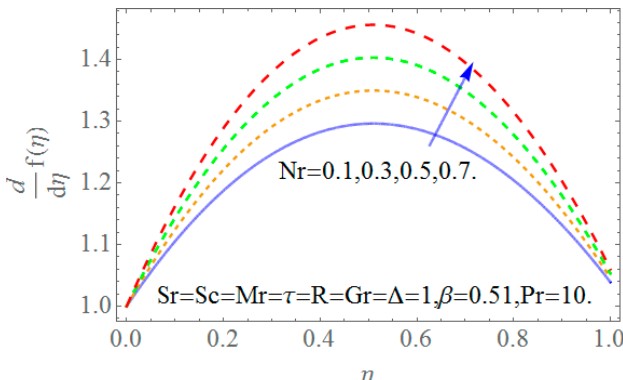

**Figure 7.** The comparison of dimensionless velocity with inertia coefficient parameter *Nr*.

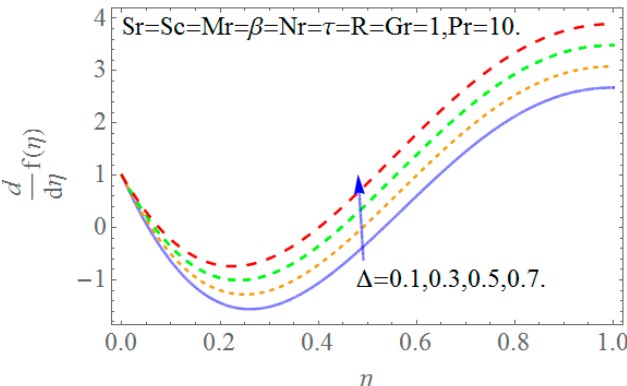

**Figure 8.** Velocity verses vortex–viscosity parameter Δ.

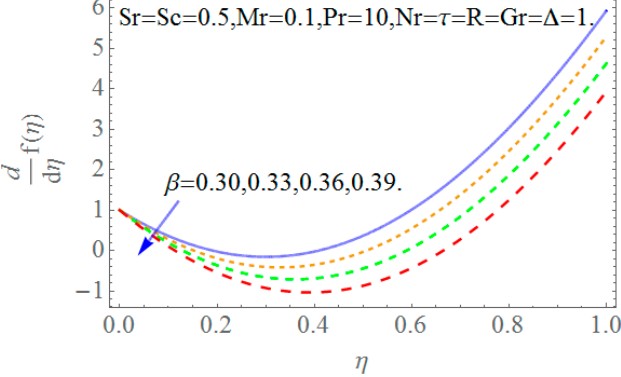

**Figure 9.** Variation of dimensionless velocity with dimensionless fluid thickness β.

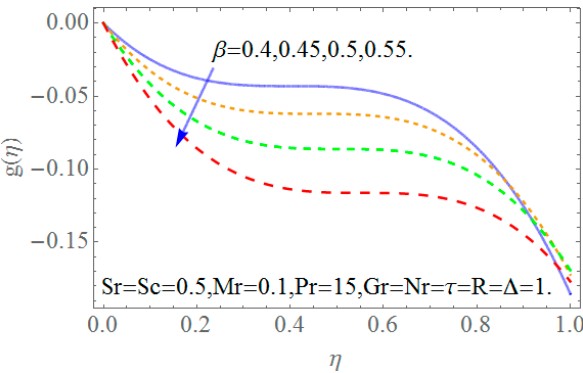

**Figure 10.** Variation of dimensionless microrotation profile with fluid thickness β.

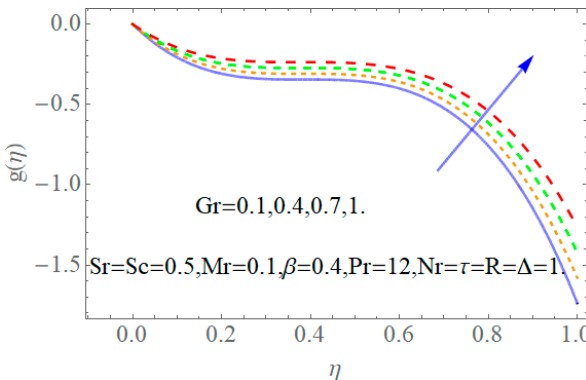

**Figure 11.** Microrotation profile under the effect of microrotation parameter *Gr*.

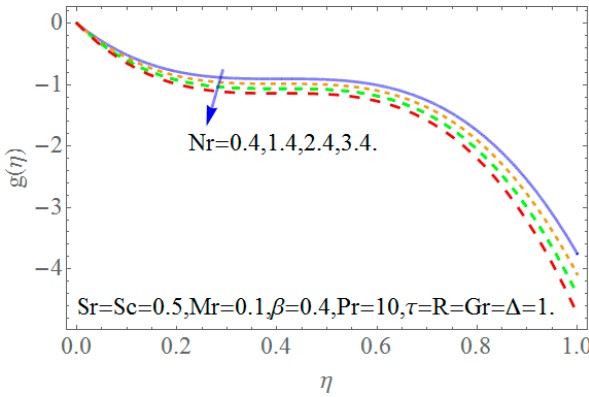

**Figure 12.** Variation of dimensionless microrotation profile with inertial parameter *Nr*.

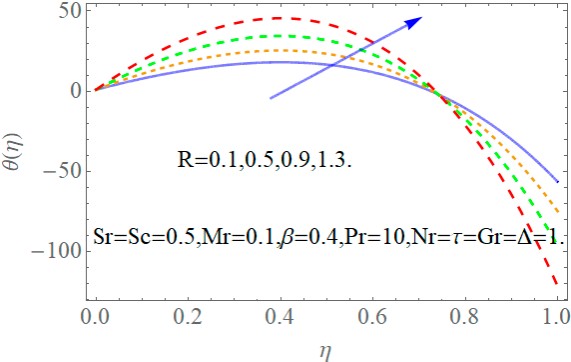

**Figure 13.** Temperature verses radiation parameter *R*.

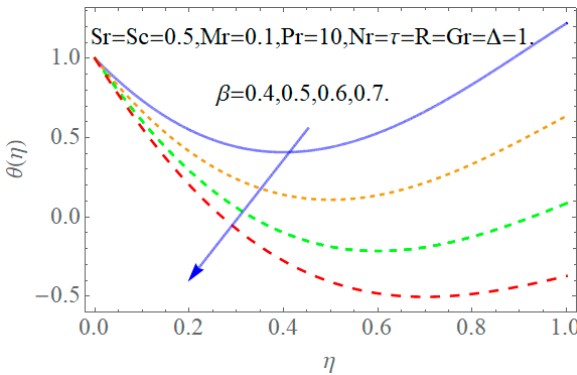

**Figure 14.** Temperature verses film thickness parameter β.

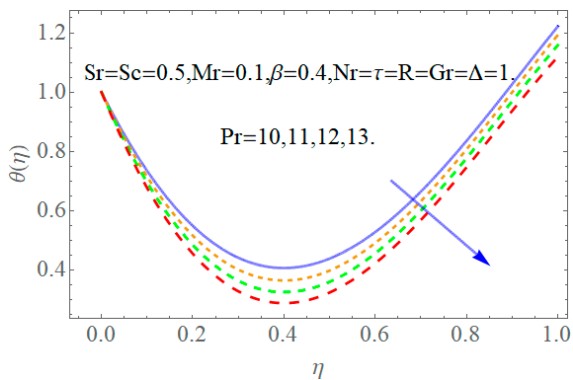

**Figure 15.** Temperature versus Prandtl number *Pr*.

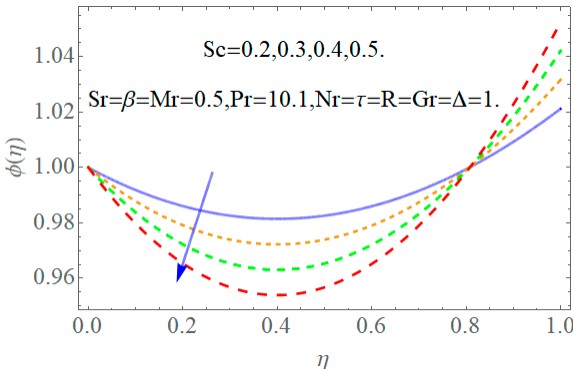

**Figure 16.** Variation of dimensionless concentration with Schmidt number *Sc*.

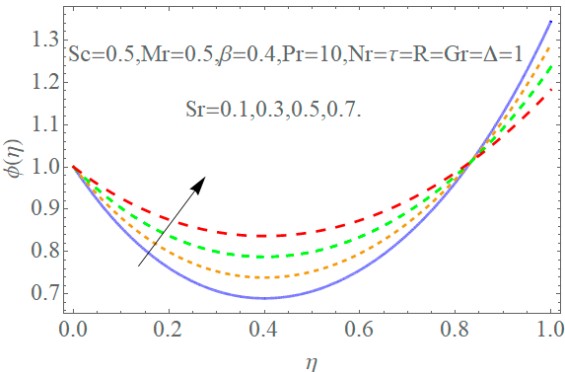

**Figure 17.** Variation of dimensionless concentration with Soret number *Sr*.

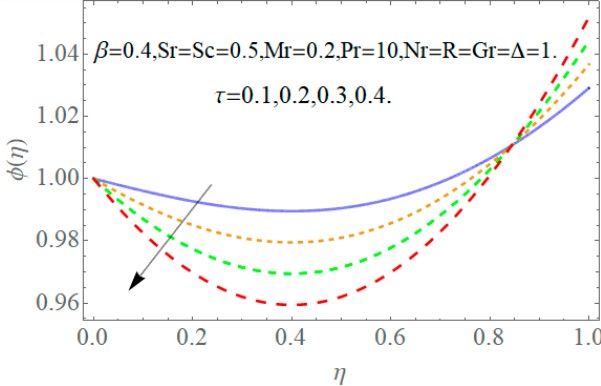

**Figure 18.** Concentration versus thermophoretic parameter τ.

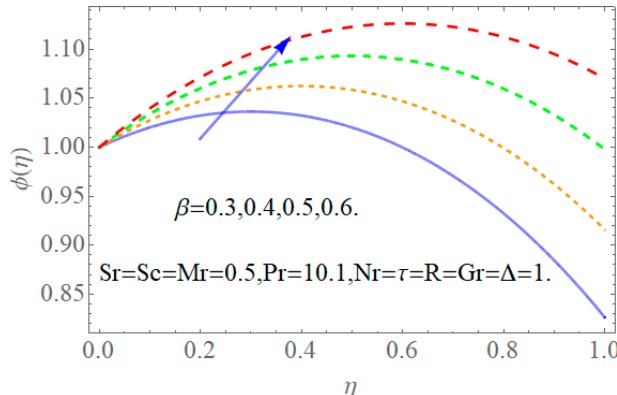

**Figure 19.** Variation of dimensionless concentration with dimensionless fluid thickness β.

## 5. Conclusions

The study of the thin film flow in a permeable medium past a stretched plate was examined. The micropolar fluid was used as a base fluid with the influence of thermal radiation and thermophoresis. Modeled non-linear coupled differential equations were tackled through HAM. The HAM solution was compared with the numerical method and close agreement was observed for the validation of the problem. The effects of the physical parameters on the velocity, temperature, and concentration profiles were displayed and discussed.

The outcomes of the problem are pointed out as follows:

- The increasing values of the thin film thickness parameter β improve the resistance force to decline the velocity and microrotation profiles, and enhance the concentration field.
- It was observed that the rise in the Soret number *Sr* enhances the concentration field $\phi(\eta)$.
- The temperature field rises with the increasing value of the thermal radiation parameter *R* because of the rate of energy and transport growth, and consequently enhances the temperature profile.
- The increase in the thickness of the thin film β reduces the temperature profile. Physically, heat transfer is larger in the thin film as compared with the thick film, while the concentration field increases as the thin film parameter β increases.
- The larger vortex–viscosity parameter Δ causes the velocity of the liquid film to rise.
- The HAM solution was validated with the numerical solution (ND-solve) and very close agreement was observed.

**Author Contributions:** V.A. and T.G. modelled the problem and drew the physical sketch. S.A. and F.A. introduced the similarity transformation and transformed the modeled problem into dimensionless form. S.O.A. solved the problem numerically and computed the results. I.K. discussed the results with conclusions. All the authors equally contributed in writing and revising the manuscript.

**Funding:** The authors would like to thank Deanship of Scientific Research, Majmaah University for supporting this work under the No. 1440-25.

**Conflicts of Interest:** The authors declare no conflict of interest.

## Nomenclature

| | |
|---|---|
| $x, y$ | Cartesian coordinates |
| $u, v$ | Velocity components |
| $U_w$ | Stretching velocity |
| $\delta$ | Uniform thickness of the thin film |
| $T_w$ | Wall temperature field |
| $C_w$ | Surface concentration |
| $T_{ref}$ | Reference temperature |
| $C_{ref}$ | Reference concentration |
| $v$ | Kinematic viscosity |
| $\mu$ | Dynamic viscosity |
| $S$ | constant characteristic |
| $C_r$ | Forchheimer inertia constant |
| $k_c$ | coupling constant |
| $T$ | Temperature field |
| $C$ | Concentration field |
| $\rho$ | Fluid density |
| $h(t)$ | Liquid film thickness |
| $q_r$ | Radiative heat fluctuation |
| $\sigma$ | Stefan–Boltzmann constant |
| $D_m$ | Concentration molecular diffusivity |
| $T_m$ | Mean temperature |
| $K$ | permeability |
| $\psi$ | Stream function |
| $\beta$ | Non-dimensional thickness of the Nano liquid film |
| $\varphi$ | porosity parameter |
| $Pr$ | Prandtl number |
| $Sc$ | Schmidt number |
| $Sr$ | Soret number |
| $G_1$ | is the microrotation constant |
| $R$ | Thermal radiation parameter |
| $V_T$ | Thermophoretic velocity |

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
