# Peer review of "Thin Film Flow of Micropolar Fluid in a Permeable Medium"

_coatings, doi:10.3390/coatings9020098_

Reviewer 1 Report

Please see the attachment file.

Reviewer 2 Report

The manuscirpt of Thin film flow of Micropolar fluid in a permeable medium is interesting article.

I commented as follows;

1.(major)

The author used many dimensionless parameters.

Definition is not clear.

The author should explain the details of them deeply.

2.(major)

Equation (19) is broken.

The author should revise.

3.(major)

All figures are very low resolutions.

The author should revise.

Please check after generation of PDF file.

Reviewer 3 Report

This manuscript addresses a numerical work describing the flow of a micropolar fluid in a porous medium under the influence of thermophoresis.

While the subject is interesting and very much in line with the topic of this journal there are several issues that must be cleared out in the manuscript before it is suitable for publication. The presentation of the model is a little bit disorganized. The authors mainly throw the equations without introducing the model. The geometry is also introduced very late in the paper. Furthermore it is not perfectly clear which are the practical conditions that are being addressed here, i.e. what is the actual situation for which the model can be applied. Finally there are some sentences in the discussion which should be clarified. In line with this I recommend the manuscript to be revised, especially considering these comments, together with the specific comments/questions, listed as follows:

In the abstract the authors state that the modeling of the temperature field is quite different from the published in the works cited. I suggest to rephrase it and refer the original approach used, specifying which are the main differences from what is already existing in the literature.

The first paragraph of the Introduction is a little bit misleading. Micropolar fluids are mixed with non-Newtonian fluids, when it should be clearer to provide first a simple definition of micropolar fluids. Then the authors may specify particular cases. Later, in the end of the Introduction the authors refer that in references [22,24,27], when the radiation parameter is set to zero, energy equation becomes meaningless. Please explain why that happens. The readers do not have to be familiar with the models in the cited references.

Mathematical formulation: Page 2, line 89, when the authors refer that the fluid is considered to be dark, I think they mean that it is approximated to a black body?

It would help the readers if the authors would introduce the equations and explain the specific integration of the equations and of the boundary conditions, so the readers may follow more easily what is being modeled. Mass, momentum and energy (with the rotation) are all mixed up and immediately simplified without explaining the geometry and “computational domain”. Fig.1 should be presented already when the model is introduced (in lines 85-91), to help introducing and clarifying the model. Why is radiation only considered in the x direction?

Page 3, line 112: “The flux is assumed small” – how small? Please quantify, provide an order of magnitude.

What is the physical reason for the radiation to have such an important role in this kind of flow?

In line with my previous comment, how was this model validated?

Page 11, line 273-274: when the authors refer to the relation between the boundary layer thickness and the thermal radiation, they are referring to the hydrodynamic boundary layer When the reason for this relation is explained in the following lines (lines 276-281) the sentence is a little bit confusing, I think it is a problem of the English language. Size of the thin film is ambiguous, the authors are just referring to the thickness right? Also the thickness of the boundary layer decreases because it cools down…ok, but why? It is probably due to viscous effects? Also it is not the increase of the Prandtl number that causes the thermal boundary layer to decrease. The increase of the Prandtl number is indicative or can be interpreted as the thickness of the hydrodynamic boundary layer to be larger than that of the thermal boundary layer, or that the viscous diffusion is larger than the thermal diffusion. So please reorganize the discussion and this kind of sentences to be scientifically more precise and focus on the physical description of the phenomena. The problem is mainly with the English and not with the scientific merit of the work, which is worth of attention. It is mainly to improve the way the sentences are written and explain the phenomena in a more physical way.

Page 12, lines 287-288: It is experimentally proved that larger values of tau lead to the reduction in mass rate. Please sustain this sentence with some citation(s). This is not proven by your work, so it must be sustained by the work cited in the literature.

How did you select the ranges for your parametric studies (to vary the viscosity parameter, concentration, Soret number, etc?

In the conclusions, please stress the physical importance of the new inputs you introduced in the model, related to the radiation parameter (why is it important/relevant to describe the physics governing the flow).

Please revise the entire text. There are sentences which are not very clear. For instance: “The idea of mass transfer in this phenomenon”, I believe that the authors mean that the mass transfer mechanisms in the process of removing the particles”…The expression”The idea of is” is not clear at all. Also there are several minor issues like typos, agreement of the verbs with the subjects (signlular vs plural). Minor corrections, which nevertheless should be done as some sentences become difficult to understand.

Author Response

Round  2

Reviewer 1 Report

It is no comment!

Author Response

Dear Referee authors are thankful to you for your kind suggestions and then finally accepting our article.

Reviewer 2 Report

The revised manuscript is satisfied as my comments.

I recommend the acceptance for publication after minor revisions.

I commented as follows;

1.(minor)

Line 359. 5[.Conclusions is mis-spelled.

The author should be revise.

2.(minor)

The many letters and symbols is used.

The author should summarize them as Nomenclatures.

Author Response

Dear Referee authors are thankful to you for your kind suggestions and then finally accepting our article.

Referee#2

The revised manuscript is satisfied as my comments.

I recommend the acceptance for publication after minor revisions.

I commented as follows;

1.(minor)

Line 359. 5[. Conclusions is mis-spelled.

The author should be revise.

Answer: (Rectified as suggested)

2.(minor)

The many letters and symbols is used.

The author should summarize them as Nomenclatures.

Answer: summarized in the revised manuscript as reference [37]

Reviewer 3 Report

The authors revised the manuscript taking into account most of the comments and concerns of the reviewers. In line with this I now consider the manuscript is suitable for publication.

However, I could still find some typos along the text, I suggest here minor amendments. Probably a careful review in the stage of paper production takes care of these minor details:

Page 2, lines 58-60: He clarified that the Navier-Stokes equation is not sufficient to handle the Cauchy stress tensor of micropolar fluid and therefore, this fluid belongs to non-Newtonian fluids.

Please correct the minor errors, namely Navier-Stokes and is not sufficient (The verb is missing).

Page 2 lines 69-70: Instead of  saying “The idea of heat and mass transfer…”, I would suggest “Heat and mass transfer mechanisms were described by the researchers”

Page 13 line 307: I suggest to say “In fact the larger Soret number” and then in line 308 I suggest to say “Fig 18 shows…”

Author Response

Referee#3

Comments and Suggestions for Authors

The authors revised the manuscript taking into account most of the comments and concerns of the reviewers. In line with this I now consider the manuscript is suitable for publication.

However, I could still find some typos along the text, I suggest here minor amendments. Probably a careful review in the stage of paper production takes care of these minor details:

Page 2, lines 58-60: He clarified that the Navier-Stokes equation is not sufficient to handle the Cauchy stress tensor of micropolar fluid. Therefore, micropolar fluid belongs to non-Newtonian fluids.

Please correct the minor errors, namely Navier-Stokes and is not sufficient (The verb is missing).

Answer: (Rectified as suggested)

Page 2 lines 69-70: Instead of saying “The idea of heat and mass transfer…”, I would suggest “Heat and mass transfer mechanisms were described by the researchers”

Answer: (Rectified as suggested)

Page 13 line 307: I suggest to say “In fact the larger Soret number” and then in line 308 I suggest to say “Fig 18 shows…

Answer: (Rectified as suggested)

     Thanks to the honorable reviewers for their suggestions